# ENTROFORMER: A TRANSFORMER-BASED ENTROPY MODEL FOR LEARNED IMAGE COMPRESSION

**Yichen Qian**
Alibaba Group
Hangzhou, China
yichen.qyc@alibaba-inc.com

**Ming Lin**
Alibaba Group
Bellevue, WA, 98004, USA
ming.l@alibaba-inc.com

**Xiuyu Sun**[*]
Alibaba Group
Hangzhou, China
xiuyu.sxy@alibaba-inc.com

**Zhiyu Tan**
Alibaba Group
Hangzhou, China
zhiyu.tzy@alibaba-inc.com

**Rong Jin**
Alibaba Group
Bellevue, WA, 98004, USA
jinrong.jr@alibaba-inc.com

## ABSTRACT

One critical component in lossy deep image compression is the entropy model, which predicts the probability distribution of the quantized latent representation in the encoding and decoding modules. Previous works build entropy models upon convolutional neural networks which are inefficient in capturing global dependencies. In this work, we propose a novel transformer-based entropy model, termed *Entroformer*, to capture long-range dependencies in probability distribution estimation effectively and efficiently. Different from vision transformers in image classification, the Entroformer is highly optimized for image compression, including a top-$k$ self-attention and a diamond relative position encoding. Meanwhile, we further expand this architecture with a parallel bidirectional context model to speed up the decoding process. The experiments show that the Entroformer achieves state-of-the-art performance on image compression while being time-efficient. Code is available at https://github.com/damo-cv/entroformer.

## 1 INTRODUCTION

Image compression is a fundamental research field in computer vision. With the development of deep learning, learned methods have led to several breakthroughs in this task. Currently, the state-of-the-art (SOTA) deep image compression models are built on the auto-encoder framework (Hinton & Salakhutdinov, 2006) with an entropy-constrained bottleneck (Theis et al., 2017; Ballé et al., 2017; Ballé et al., 2018; Mentzer et al., 2018; Minnen et al., 2018a; Lee et al., 2019; Guo et al., 2021; Sun et al., 2021). An entropy model estimates the conditional probability distribution of latents for compression by standard entropy coding algorithms. In this paper, we focus on improving the predictive ability of the entropy model, which leads to higher compression rates without increasing distortion.

One of the main challenges of the entropy model is *how to learn representation for various dependencies*. For example, as shown in Figure 1, the edges have long-range dependencies; and the five hats are likely to be related in shape; the region of the sky has identical color. Some works use local context (Minnen et al., 2018a; Lee et al., 2019; Mentzer et al., 2018) or additional side information (Ballé et al., 2018; Hu et al., 2020; Minnen et al., 2018b) for short-range spatial dependencies, while others use non-local mechanism (Li et al., 2020; Qian et al., 2021; Cheng et al., 2020; Chen et al., 2021) to capture long-range spatial dependencies. However, the constraint of capturing long-range spatial dependencies still remains in these CNN-based methods.

Another main challenge is *the trade-off between performance and decoding speed*. Though previous context based method uses unidirectional context model to improve the predictive ability, it suffers

---

[*]Corresponding author

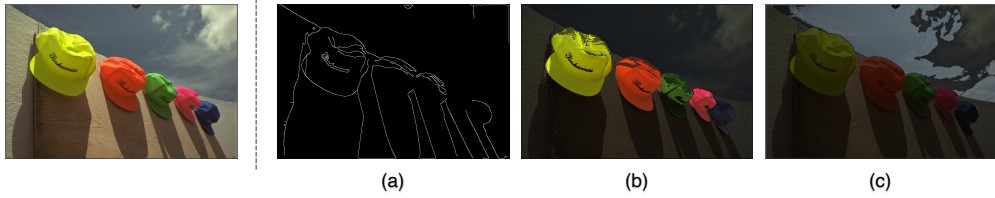

Figure 1: Dependencies exist in (a) texture, (b) semantic and (c) color space.

from slow decoding speed. It decodes symbols in a raster-scan order with $O(n)$ serial process that can not be accelerated by modern GPUs. A two-pass parallel context model (He et al., 2021) is introduced for acceleration, which decodes symbols in a particular order to minimize serial processing. However, this parallel context model uses a weak context information, which degrades the compression performance.

In this paper, we propose a novel transformer-based entropy model termed as *Entroformer* to address the above two challenges in one shot. Following the Transformer (Vaswani et al., 2017), self-attention is used to relate different positions of a single latent in order to compute a representation of the latents. In Entroformer, spatial and content based dependencies are jointly taken into account in both hyperprior and context model. To further optimize Entroformer for image compression, the following designs are innovated: (1) Entroformer uses a multi-head attention with a top-$k$ selection to distill information in representation subspaces: The multi-head attention provides a learning-based partition to learn dependencies in different views; and the top-$k$ scheme filters noisy signals by select the most similar $k$ latents in the self-attention module, which is crucial for the convergence of Entroformer. (2) To inherit the local bias of CNNs, a novel position encoding unit is designed to provide better spatial representation for the image compression. This position encoding is based on a relative position encoding with a diamond-shape boundary (Diamond RPE), which embeds $\ell_1$-norm distance prior knowledge in image compression. (3) The two-pass decoding framework (He et al., 2021) is utilized to speed up the decoding of Entroformer. A bidirectional context with long-range context is introduced instead of the local checkeboard context, which helps counteract the performance degradation. In summary, our contributions include:

- In this paper, Entroformer is proposed to improve learned image compression. To our knowledge, it is the first successful attempt to introduce Transformer based method to image compression task. Experiments show that our method outperforms the most advanced CNNs methods by 5.2% and standard codec BPG (Bellard., 2014) by 20.5% at low bit rates.

- To further optimize Entroformer for image compression, the multi-head scheme with top-$k$ selection is proposed to capture precise dependencies for probability distribution estimation, and the diamond relative position encoding (diamond RPE) is proposed for better positional encoding. Experiments show that these methods support image compression task to keep training stable and obtain superior results.

- With the help of bidirectional context and two-pass decoding framework, our parallel Entroformer is more time-efficient than the serialized one on modern GPU devices without performance degradation.

## 2 COMPRESSION WITH HYPERPRIOR AND CONTEXT

There are types of entropy models such as hyperprior (Minnen et al., 2018b; Ballé et al., 2018), context model (Li et al., 2018; Mentzer et al., 2018), and the combined method (Lee et al., 2019; Minnen et al., 2018a; Minnen & Singh, 2020). Hyperprior methods typically make use of side information of the quantized latent representation with additional bits. Context models learn auto-regressive priors incorporating prediction from the causal context of the latents, which is a bit-free. Our Entroformer combines a context model with hyperprior based on previous works (Minnen et al., 2018b; Ballé et al., 2018).

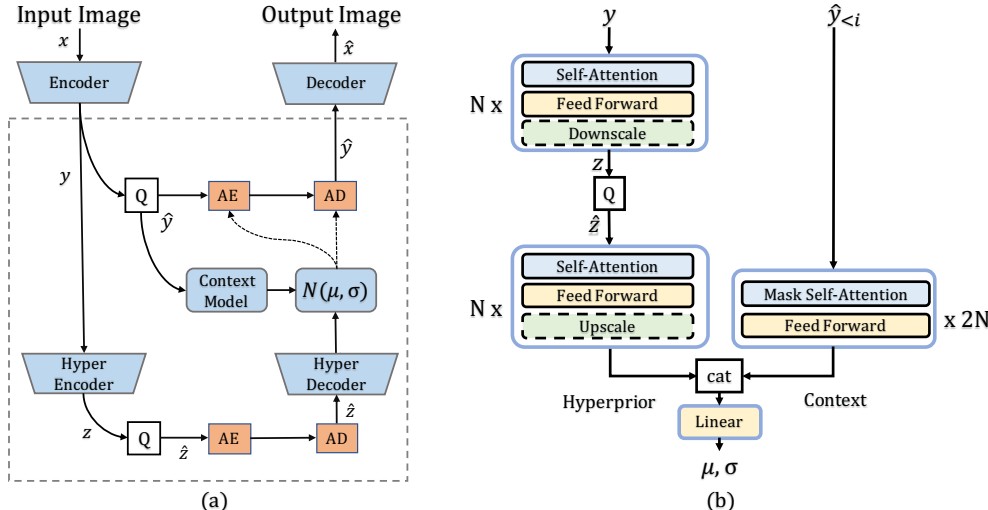

Figure 2: In (a), we show the architecture of image compression model with with hyerprprior and context model. The main autoencoder learns an latent representation of image, while a hyper-autoencoder learns a hyperprior representation. Context model learns correlation of latents from their decoded context. Real-valued latent representation $(y, z)$ is quantized ($Q$) to create latents ($\hat{y}$) and hyper-latents ($\hat{z}$), which are compressed into a bitstream using an arithmetic encoder ($AE$) and decompressed by an arithmetic decoder ($AD$). The latents use a Gaussian entropy model conditioned on hyperprior and context. The region in dashed line corresponds to our Entroformer, which is shown in (b). A transformer encoder learns the hyerprprior, while a transformer decoder learns the context using masked self-attention. Linear component is used to joint these features to generate the Gaussian parameters (*i.e.* $\mu, \sigma$).

We describe the architecture of compression model with Entroformer in Figure 2. The main autoencoder learns a quantized latent representation $\hat{y}$ of image $x$. The reconstructed image is denoted as $\hat{x}$. The hyper-autoencoder learns a quantized hyper-latents representation $\hat{z}$. Following the work of Ballé et al. (2016), the quantization operation is approximated by an additive uniform noise during training. This ensures a good match between encoder and decoder distributions of both the quantized latents, and continuous-valued latents subjected to additive uniform noise during training. Following the work of Minnen et al. (2018a), we model each latent, $\hat{y}_i$, as a Gaussian with mean $\mu_i$ and deviation $\sigma_i$ convolved with a unit uniform distribution:

$$p_{\hat{y}}(\hat{y}|\hat{z}, \theta) = \prod_{i=1}(\mathcal{N}(\mu_i, \sigma_i^2) * \mathcal{U}(-0.5, 0.5)))\,(\hat{y}_i) \tag{1}$$

where $\mu$ and $\sigma$ are predicted by the entropy model, $\theta$ is the parameters of the entropy model. The entropy model consists of hyper-autoencoder, context model, and parameter networks. Since we do not make any assumptions about the distribution of the hyper-latents, a non-parametric, fully factorized density model is used.

The training goal for learned image compression is to optimize the trade-off between the estimated coding length of the bitstream and the quality of the reconstruction, which is a rate-distortion optimization problem:

$$\mathcal{L} = R + \lambda D = \underbrace{\mathbb{E}_{x \sim p_x}[-\log_2 p_{\hat{y}}(\hat{y})]}_{\text{rate (latents)}} + \underbrace{\mathbb{E}_{x \sim p_x}[-\log_2 p_{\hat{z}}(\hat{z})]}_{\text{rate (hyper-latents)}} + \underbrace{\lambda \cdot \mathbb{E}_{x \sim p_x}||x - \hat{x}||_2^2}_{\text{distortion}}, \tag{2}$$

where $\lambda$ is the coefficient which controls the rate-distortion trade-off, $p_x$ is the unknown distribution of natural images. $p_{\hat{z}}$ use a fully factorized density model. The first term represents the estimated compression rate of the latent representation, while the second term represents the estimated compression rate of the hyper-latent representation. The third term represents the distortion value under given metric, such as mean squared error (MSE).

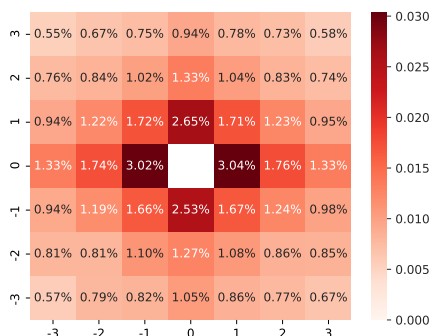

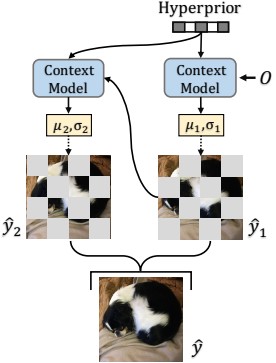

Figure 3: Rate increase ratios of different position when the corresponding context is masked. Closer context contributes to more bitrate influence, whose trend follows a particular pattern of diamond shape.

Figure 4: Parallel bidirectional context model. The first slice ($\hat{y}_1$) is conditioned solely on the hyperprior. The second slice ($\hat{y}_2$) is conditioned on the hyperprior and the decoded context (*i.e.* first slice).

## 3 TRANSFORMER-BASED ENTROPY MODEL

Our model contains two main components, a CNN-based autoencoder to learn a latent representation, a transformer-based entropy model to predict latents. We describe our architecture in Figure 2 (a), and our Entroformer in detail in Figure 2 (b). In this section, we first propose two ingredients, a diamond relative position encoding (diamond RPE) and a top-k scheme, which are essential for image compression. Then, we extend the checkboard context model (He et al., 2021) to a parallel bidirectional context model.

### 3.1 TRANSFORMER ARCHITECTURE

In entropy model design we follow the original Transformer (Vaswani et al., 2017), which employs an encoder-decoder structure. We model the latent representation $\hat{y}$ as a sequence. The latents sequence is then mapped to $D$ dimensions with a trainable linear projection. For hyperprior, we stack $N$ transformer encoder layers to yield hyper-latent $z$, which is quantized and fed to $N$ transformer-encoder layers to generate hyperprior. To produce a hierarchical representation, the resolution of feature is changed by downsample and upscale module. For autoregressive prior, we stack $2N$ transformer decoder layers to generate autoregressive features. To generate the Gaussian parameters, a linear layer is attached to the combination of hyperprior features and context features.

For any sequence of length $n$, the vanilla attention in the transformer is the dot product attention (Vaswani et al., 2017). Following the standard notation, we compute the matrix of outputs via the linear projection of the input matrix $X \in \mathbb{R}^{n \times d_m}$:

$$\text{Attention}(X) = \text{softmax}\left(\frac{XW^Q(XW^K)^T}{\sqrt{d_k}}\right) XW^V \tag{3}$$

$W^Q, W^K \in \mathbb{R}^{d_m \times d_k}$ and $W^V \in \mathbb{R}^{d_m \times d_v}$ are learned parameter matrices.

### 3.2 POSITION ENCODING

To explore the impact of position in image compression, we first design a transformer-based entropy model without any position encoding. We feed a random mask to self-attention during training. During testing, we evaluate the effect of each position $i$ by employing a corresponding mask, which are set to 1 except position $i$. Figure 3 plots the impact in bit rate of each position, compared to the result with context of all positions. This result highlights how the rate effected by the position of context. This observation presents a comprehensive understanding and provides empirical guideline for new position encoding design.

In order for spatial information of the latents, one common approach is to use biased attention weights based on relative relation (Shaw et al., 2018). Based on the result in Figure 3, we propose an extension to this relation-aware self-attention to consider the element position influence on image compression performance.

Each attention operates on an input sequence, $x = (x_1, ..., x_n)$ of $n$ elements where $x_i \in \mathbb{R}^{d_m}$. The edge between input elements $x_i$ and $x_j$ is represented by vectors $p_{ij}^K \in \mathbb{R}^{d_k}$. We modify the dot-product attention matrix in eq. 3 to a compatibility function that compares two input elements in relation-aware self-attention:

$$e_{ij} = \frac{x_i W^Q (x_j W^K + p_{ij}^K)^T}{\sqrt{d_k}} \tag{4}$$

The weight coefficient, $A$, is computed using a softmax on dot-product attention matrix.

On image plane, the relative position $a_{i-j}$ is a 2D coordinate. And we modify the maximum absolute relative position to a 2D boundary with diamond shape. As shown in Figure 3, we observe that the closer contexts latent save more bit rate in context model. That tells how the distance of context latents influences the bit saving of context modeling in learned image compression. Therefore, we consider a 2D boundary with a maximum $\ell_1$ distance value of $h$.

$$p_{ij}^K = w_{clip(a_{i-j}, h)}^K$$
$$clip(a_{i-j}, h) = \begin{cases} a_{i-j} & \|a_{i-j}\|_1 <= h \\ (h, h) & \text{otherwise.} \end{cases} \tag{5}$$

We then learn the relative position encoding $w^K = (w_{-h,-h}^K, w_{-h,-h+1}^K, ..., w_{h,h-1}^K, w_{h,h}^K)$.

### 3.3 TOP-k SCHEME IN SELF-ATTENTION

The vanilla attention in the transformer is the scaled dot-product attention. It computes the dot products of the query with all keys, divide each by $\sqrt{d_k}$, and apply a softmax function to obtain the weights on the values.

Content-based sparse attention methods (Wang et al., 2021; 2022) have demonstrated promising results on vision tasks. Inspired by these works, we select the top-k most similar keys for each query instead of computing the attention matrix for all the query-key pairs as in vanilla attention. The queries, keys and values are packed together into matrices $Q \in \mathbb{R}^{n \times d_k}$, $K \in \mathbb{R}^{n \times d_k}$ and $V \in \mathbb{R}^{n \times d_v}$. $P \in \mathbb{R}^{n \times d_k}$ denotes relative position encodings described in Sec 3.2. The row-wise top-k elements in attention matrix are selected for softmax function:

$$V' = \text{softmax}\left( f_k \left( \frac{Q(K+P)^T}{\sqrt{d_k}} \right) \right) V \tag{6}$$

where $f_k(\cdot)$ denotes the row-wise top-k selection operator:

$$f_k(e_{ij}) = \begin{cases} e_{ij} & \text{if } e_{ij} \text{ is within the top-k largest elements in row } j \\ -\infty & \text{otherwise.} \end{cases} \tag{7}$$

Albeit easing sequence length mismatching problem by relative position encoding, transformer-based entropy model also suffers from token quantity mismatching in self-attention. For extremely larger sequence length during inference than training, a large quantity of irrelevant context latents would push the relevant ones to a small weight. To counteract this effect, we modify the self-attention with top-k scheme in the Entroformer. It helps to distill irrelevant context latents and stable training procedure.

### 3.4 PARALLEL BIDIRECTIONAL CONTEXT MODEL

The transformer architecture can be used to efficiently train autoregressive model by masking the attention computation such that the $i$-th position can only be influenced by a position $j$ if and only if $j < i$. This causal masking uses a limited unidirectional context, and makes autoregressive model impossible to be parallel.

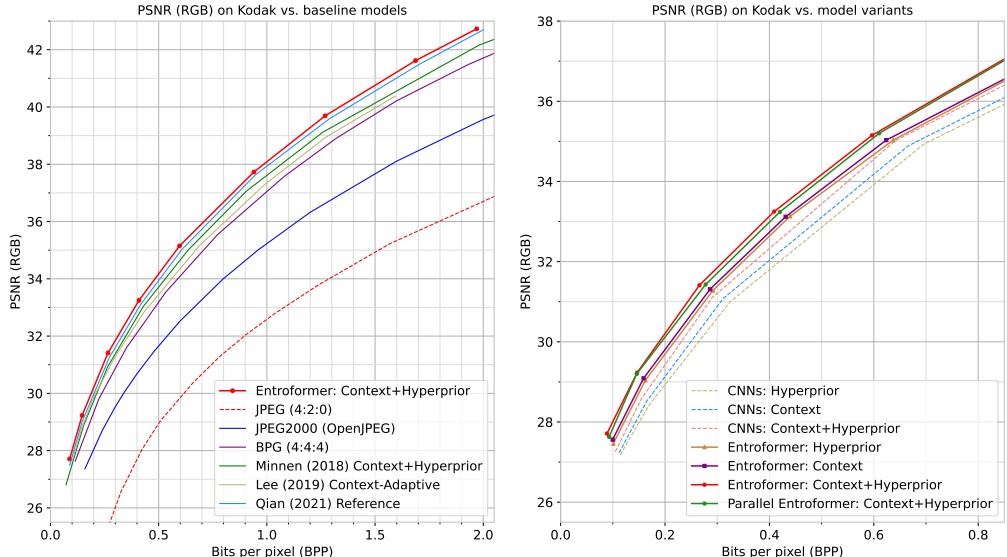

Figure 5: The Entroformer (context + hyperprior) achieves better RD performance than previous CNNs-based methods and standard codecs on the Kodak image set as measured by PSNR (RGB) (left). The right graph compares the relative performance of different versions of our method. Our Entroformer architecture helps significantly at hyperprior-only and context-only model by 12.4% and 11.3% respectively at low bit rates. Note that parallel Entroformer has almost no effect ( about 1% bpp increase) for RD performance.

Following the work of He et al. (2021), the latents are split into two slices along the spatial dimension. Figure 4 provides a high-level overview of this architecture. The decoding of the first slice ($\hat{y}_1$) is run in parallel by using a Gaussian entropy model conditioned solely on the hyperprior. Then the decoded latents become visible for decoding the second slice ($\hat{y}_2$) in the next pass. Since the context are accessible for each latent in second slice, the second slice can be processed in parallel. The decoding of the second slice is also run in parallel conditioned on both the hyperprior and the decoded context latents (*i.e.* first slice). Therefore, the decoding of this structure can be performed in two serial processing.

When decoding the second slice, we adjust the mask in the self-attention module to build a bidirectional context model, which introduces the future context to benefit accurate prediction. For this reason, though the performance of the first slice degrades (due to hyperprior only), Entroformer counteracts this effect by providing a more powerful context model to promote the performance of the second slice. In the other hand, compared to the CNN-based accelerated method (about 4% performance degradation) (He et al., 2021), our transformer-based method can utilize rich context and achieve a better balance between speed and performance (about 1% performance degradation).

## 4 EXPERIMENTAL RESULTS

We evaluate the effects of our transformer-baed entropy model by calculating the rate–distortion (RD) performance. Figure 5 shows the RD curves over the publicly available Kodak dataset (Kodak, 1993) by using peak signal-to-noise ratio (PSNR) as the image quality metric. As shown in the left part, our Entroformer with joint the hyperprior module and the context module outperforms the state-of-the-art CNNs methods by 5.2% and the BPG by 20.5% at low bit rates. As shown in the right part, two variant entropy models, the hyperprior-only one and the context-only one, are evaluated to isolate the effect of Entroformer architecture. Additionally, the parallel Entroformer with bi-directional context is also competitive and is better than the previous baselines.

To evaluate the importance of different components of the Entroformer, we varied our base model in different ways, measuring the change in performance on Kodak test set. When evaluating the effects of using the diamond relative position encoding and the top-$k$ scheme, we adapt the Entro-

| Model | Position Information | Mode | bpp | $\Delta$ bpp (%) |
|---|---|---|---|---|
| CNNs | - | - | 0.6621 | - |
| Transformer | - | - | 0.6569 | 0.8 |
| Transformer | Absolute Position Encoding | 2D | 0.6563 | 0.9 |
| | Relative Position Encoding | 1D+1D | 0.6358 | 4.0 |
| Transformer | Relative Position Encoding | 2D | 0.6296 | 4.9 |
| | Relative Position Encoding | 2D+Diamond | **0.6238** | **5.8** |

Table 1: Ablation of position encoding methods. All models are optimized with $\lambda = 0.02$ (PNSR $\approx$ 35.0 on Kodak).

former with only the context module. When evaluating the effects of bidirectional context model, we employ the Entroformer with the context module and the hyperprior module.

**Architecture** The architecture of our approach extends on the work of Minnen et al. (2018b). As our focus is on the entropy model, we only replace the CNNs entropy model by proposed Entroformer (*i.e.* the main autoencoder is not changed). We used 6 transformer encoder blocks for hyperprior and 6 transformer decoder blocks for context. The outputs of these two modules are combined by a linear layer. The dimension of the latent embedding is 384. Each transformer layer has a multi-head self-attention (heads=6), and a feed-forward network (ratio=4). If not specified, we use serial Entroformer with the default parameters: $k$=32 (top-$k$ scheme), $h$=3 (diamond RPE).

**Training Details** We choose 14886 images from OpenImage (Krasin et al., 2017) as our training data. During training, all images are randomly cropped to $384 \times 384$ patches. Note that large patch size is necessary for the training of the transformer-based model. We optimize the networks using distortion terms of MSE. For each distortion type, the average bits per pixel (bpp) and the distortion PSNR, over the test set are measured for each model configurations. The detailed structure and the experimental settings are described in appendix.

### 4.1 ANALYSIS ON POSITION ENCODING

**Position Encoding Methods** We perform a component-wise analysis to study the effects of different position encodings in our Entroformer. We construct transformer-based context models with different position encoding. A CNN-based context model is implemented for comparison. As shown in Table 1, When applying absolute position encoding or non-position encoding in Entroformer, we can achieve a limited bpp saving than the baseline. If we apply the relative position encoding, it can perform better than the absolute position one ( 4.0% bpp saving *vs.* 0.9% bpp saving ). Additionally, it is crucial to extend the position encoding from 1D to 2D ( 4.9% bpp saving *vs.* 4.0% bpp saving ) . Finally, combining with the diamond boundary, we could get more bit-rate saving than the others, in particular, 5.8% bpp saved than the CNNs one.

**Clipping Distance** We evaluate the effect of varying the clipping $\ell_1$ distance, $h$, of diamond RPE. Figure 6 compares the number of $h$ by showing the relative reduction in bit rate at a single rate-point. The baseline is implemented by a transformer-based context model without position encoding. It achieves the best result when $h = 3$. The conclusion is similar as the analysis shown in Figure 3.

### 4.2 NUMBER $k$ OF TOP-$k$ SELF-ATTENTION

The impact of top-$k$ scheme is analyzed in Figure 7. The parameter $k$ specifies the involved token number of self-attention. We report the full learning bpp curves on Kodak test set for different number $k$ and not just single RD point. Additional curve of original self-attention mechanism is presented (dash) for comparison. The spatial size of latents is $24 \times 24$ during training and $32 \times 48$ during testing. Notably, the compression performance increase for $k \leq 64$, which is much smaller than the sequence length of 576 and 1536 for training and testing respectively. Moreover, the top-$k$ scheme also influences convergence speed of Entroformer. Interestingly, when increase $k$ larger than 64, we see quite different results that top-$k$ scheme has no difference with original self-attention. One hypothesis is that there is a vast amount of irrelevant information in dense self-attention for image compression. These observations reflect that removing the irrelevant tokens benefits the convergence of Entroformer training.

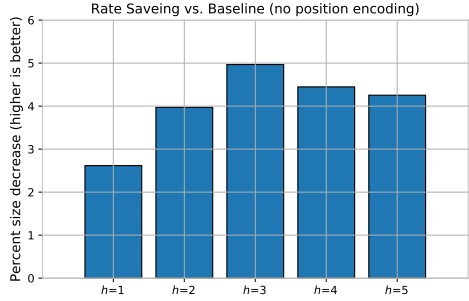 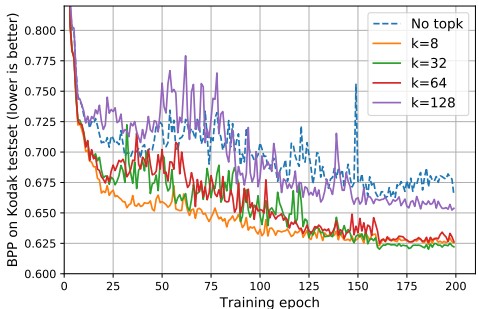

Figure 6: Results for varying the clipping $\ell_1$ distance of diamond RPE. The base implementation uses a transformer context model without position encoding. All models are optimized with $\lambda = 0.02$ (PNSR $\approx 35.0$ on Kodak).

Figure 7: Effect of restricting the self-attention to various top-$k$ similar content. All models use a transformer context model and are optimized with $\lambda = 0.02$ (PNSR $\approx 35.0$ on Kodak).

| Test set | Ballé2018 (Hyperprior) | Minne2018 (Hyperprior+Context) | Entroformer (Hyperprior+Context) | |
|---|---|---|---|---|
| | | | Serial | parallel |
| Kodak ($768 \times 512$) | 21.6 | 4229.9 | 51678.5 | 170.6 |

Table 2: Inference latency (ms) of entropy model on Kodak. Ballé et al. (2018) uses is paralleled, while Minnen et al. (2018b) is inherently serial. For a direct comparison, we test our Entroformer with serial architecture and parallel architecture.

| Half latents ($\hat{y}_2$) | Hyperprior-only | Hyperprior & unidirectional context | Hyperprior & bidirectional context |
|---|---|---|---|
| bpp | 0.3267 | 0.3105 | 0.2972 |

Table 3: Bidirectional context *vs.* unidirectional context. Combined with bidirectional context, context model achieves bit rate saving significantly. The results are test by the same model.

## 4.3 PARALLEL BIDIRECTIONAL CONTEXT MODEL

In Table 2, we report the inference latency of Entroformer (serial architecture and parallel architecture) comparing to other learned entropy models. We benchmark models' inference latency on 16GB Tesla V100 GPU card. From Table 2, we see that parallel bidirectional context model is much more time-efficient than serial unidirectional context model.

As shown in Figure 5, by employing Entroformer, performance of the parallel bidirectional context is on par with the serial unidirectional context. For checkboard parallel context model (He et al., 2021), it speeds up the decoding process at a cost of compression quality. We also study the impact of bidirectional context for compression performance on the second slice (*i.e.* $\hat{y}_2$) solely. Table 3 provides an empirical evaluation of three model variants (hyperprior-only, hyperprior with unidirectional context and hyperprior with bidirectional context). Table 3 shows that the bidirectional context is more accurate than the unidirectional context in our Entroformer.

## 4.4 VISUALIZATION

Attention mechanism in transformer is the key component which models relations between feature representations. In this section, we visualize the self-attentions mechanism of our Entroformer, focusing on a few points in the image. In Figure 8, the attention maps from different points exhibit different behaviours that seem related to structure, semantic and color respectively. It show how the Entroformer finds related context to support its distribution prediction for the current latent. In Figure 9, we visualize self-attentions heads separately from one point. We can see that different



Figure 8: Examples of the self-attention mechanism for three points.

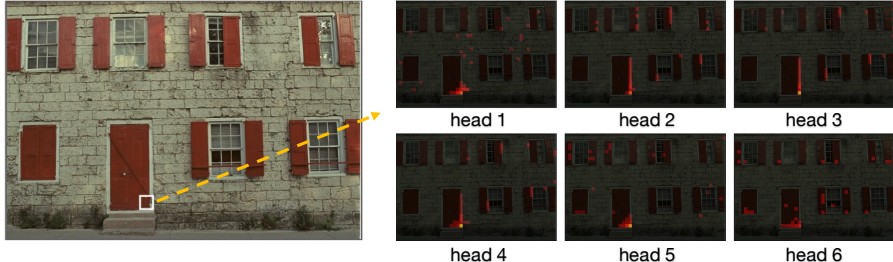

Figure 9: Self-attention heads following long-range dependencies with different patterns.

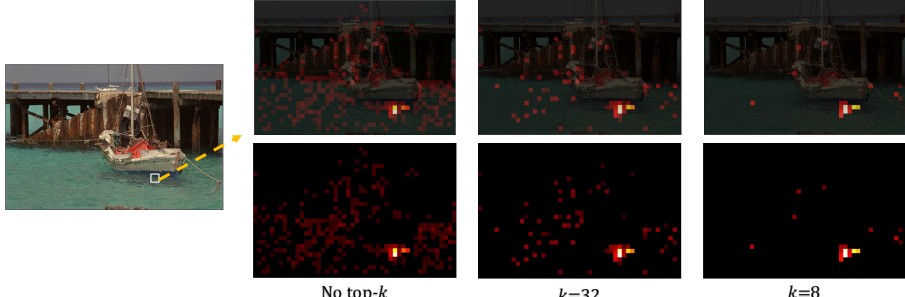

Figure 10: Self-attention from models with different $k$.

heads attend to different region of an image. This means that separate sections of the representation can learn different aspects of the meanings of each latent. In Figure 10, we visualize self-attentions heads with different $k$. Compared with dense attention, the top-$k$ attention filters out irrelevant information from background. It successfully makes the attention concentrating on the most informative context.

## 5 DISCUSSION

In this paper, Entroformer is proposed as the entropy model to improve learned image compression by accuracy probability distribution estimation. To our knowledge, this is the first successful attempt to introduce transformers in image compression task. The Entroformer follows the standard transformer framework with a novel positional encoding and top-$k$ self-attention module. The proposed diamond RPE is a 2D relative position encoding with a diamond shape boundary. The top-$k$ self-attention is helpful in dependencies distillation. In particular, the top-$k$ module benefits the convergence of neural networks training. Last but not least, the Entroformer with bidirectional context can work well in the two-pass decoding framework for fast speed without performance degradation. Experiments show that the Entroformer achieves the SOTA results in standard benchmarks.

There are still some issues to be discussed in the future. For example, (1) Whether we could build a hierarchical decoding framework to achieve a better balance between speed and RD performance, even beyond the raster-scan mode; (2) Whether the fine-grained assignment of $k$ to different images could increase performance; (3) Whether the rate control module plays a more important role in the transformer-based entropy model than in the convolution-based model. We would like to explore these issues in future work.

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

# A APPENDIX

## A.1 PSNR (RGB) ON KODAK DATASET

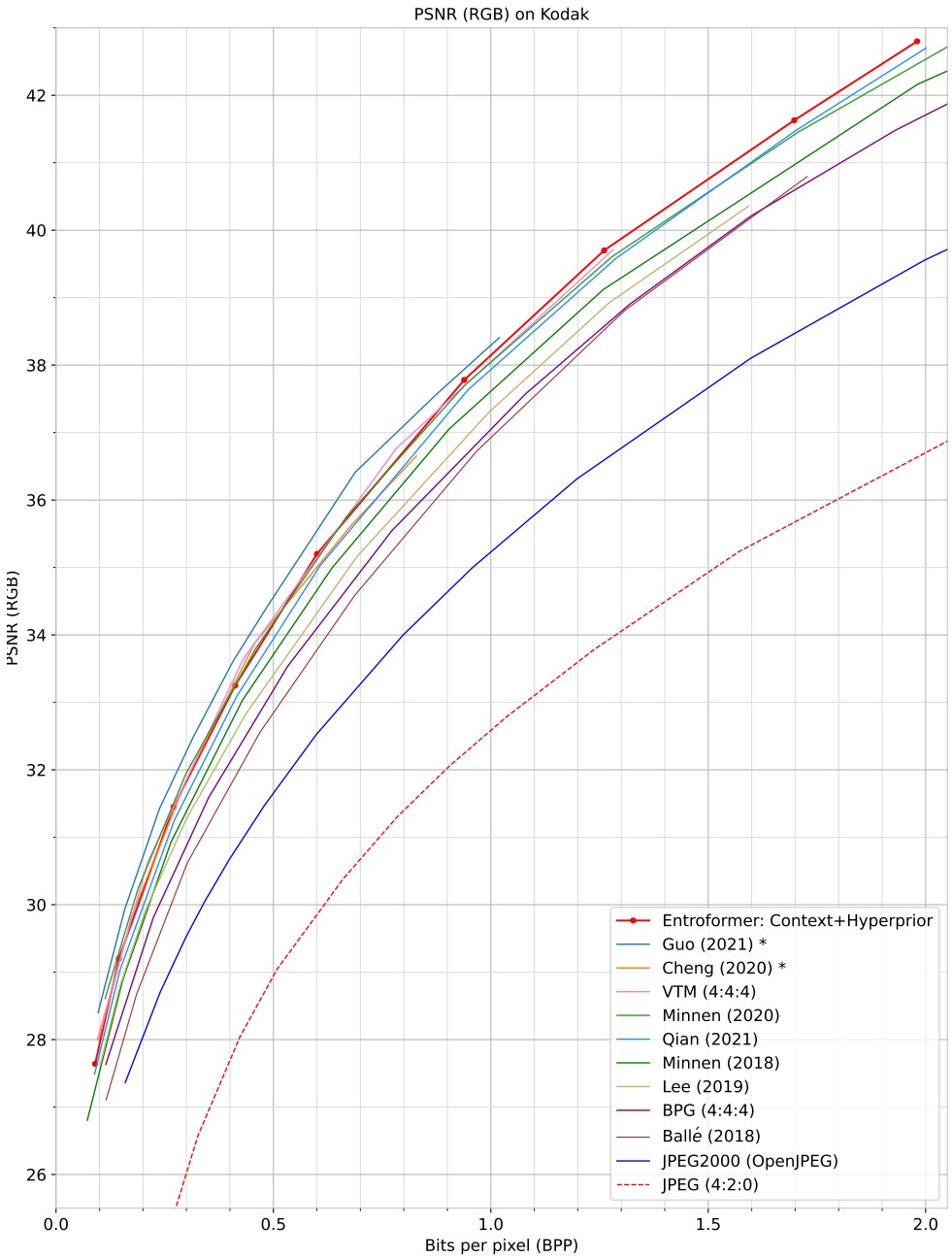

Figure 11: Rate-Distortion curves on the Kodak image set using PSNR. Each point on the RD curves is calculated by averaging over the MS-SSIM and bit rate. Note that the models with * use deeper *main auto-encoder* (transform between image and latent).

## A.2  MS-SSIM (RGB) ON KODAK DATASET

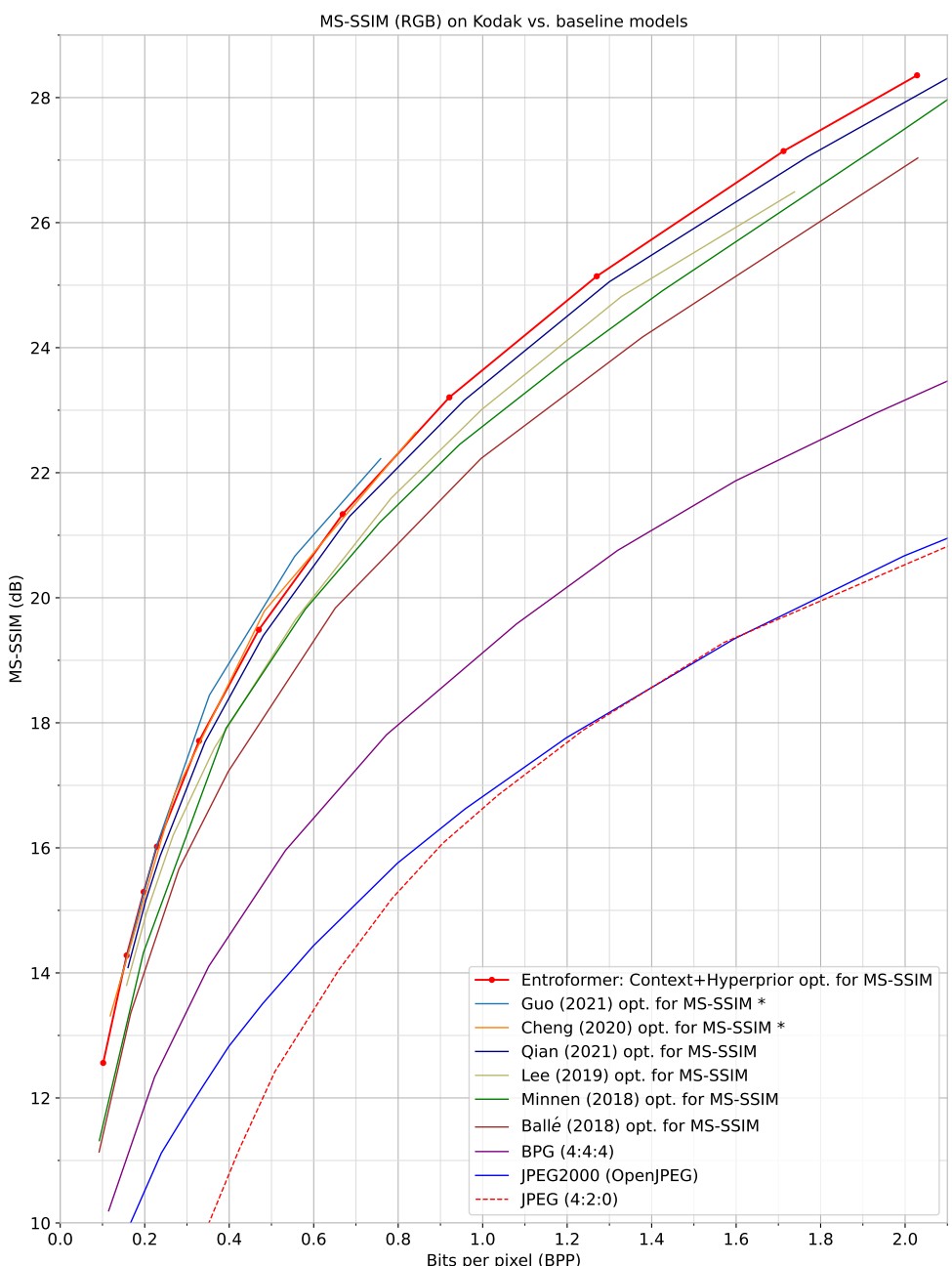

Figure 12:  Rate-Distortion curves on the Kodak image set using MS-SSIM. Each point on the RD curves is calculated by averaging over the MS-SSIM and bit rate. To improve readability, this graph shows MS-SSIM scores in dB using the formula: $\text{MS-SSIM}_{\text{dB}} = -10\log_{10}(1 - \text{MS-SSIM})$. Note that the models with $^*$ use deeper *main auto-encoder* (transform between image and latent).

## A.3   POSITION ENCODING FOR VARIOUS IMAGE SIZE

| Position Encoding | Image size | bpp | $\Delta$ bpp (%) |
|---|---|---|---|
| Absolute PE | $300 \times 300$ | 0.7654 | - |
| Relative PE | $300 \times 300$ | 0.7180 | 6.2% |
| Absolute PE | $600 \times 600$ | 0.5360 | - |
| Relative PE | $600 \times 600$ | 0.5057 | 5.7% |
| Absolute PE | $900 \times 900$ | 0.4938 | - |
| Relative PE | $900 \times 900$ | 0.4154 | 15.9% |
| Absolute PE | $1200 \times 1200$ | 0.4808 | - |
| Relative PE | $1200 \times 1200$ | 0.4099 | 14.7% |

Table 4: Performance of position encoding for varying the image size on Tecnick dataset. It shows that relative position encoding can directly generalize to larger image size. The model with absolute position encoding degrades when process larger image size than training.

## A.4   ARCHITECTURE DETAILS

| Encoder | Decoder | Hyper Encoder | Hyper Decoder | Context Model | Linear |
|---|---|---|---|---|---|
| Conv: k5c192s2 | Deconv: k5c192s2 | Trans: dim384head6 | Trans: dim384head6 | Trans: dim384head6 | Linear: dim768 |
| GDN | IGDN | Downscale | Upscale | Trans: dim384head6 | Leaky ReLU |
| Conv: k5c192s2 | Deconv: k5c192s2 | Trans: dim384head6 | Trans: dim384head6 | Trans: dim384head6 | Linear: dim768 |
| GDN | IGDN | Downscale | Upscale | Trans: dim384head6 | |
| Conv: k5c192s2 | Deconv: k5c192s2 | Trans: dim384head6 | Trans: dim384head6 | Trans: dim384head6 | |
| GDN | IGDN | | | Trans: dim384head6 | |
| Conv: k5c384s2 | Deconv: k5c192s2 | | | | |

Table 5: Each row corresponds to a layer of our generalized model. Convolutional layers are specified with the "Conv" prefix followed by the kernel size, number of output channels and downsampling stride (*e.g.* the first layer of the encoder uses $5 \times 5$ kernels with 192 output channels and a stride of 2). The "Deconv" prefix corresponds to upsampled convolutions. The "Trans" corresponds to transformer block as in Vaswani et al. (2017), followed by the inner dimension and head number. A transformer block consists of a self-attention module and a feed-forward module (the dimension of the feed-forward is 4 times of inner embedding's). GDN stands for generalized and divisive normalization, and IGDN is inverse GDN (Ballé et al., 2016). Group convolution with kernel=3 and stride=2 is used for downscale operation. Pixel shuffule (by group convolution with kernel=3 and stride=2) is used for upscale operation.

| Method | Minnen (2018) Context+Hyperprior | Minnen (2020) Channel Auto-regressive | Qian (2021) Reference | Ours |
|---|---|---|---|---|
| Weights of entropy model | 90.0M | 237.3M | 145.7M | 142.7M |

Table 6: Comparison on weight with entropy model variants.

Details about the individual network layers in each component of our models are outlined in Table 5. The output of the last layer in encoder corresponds to the latents $\hat{y}$, and the output of the last layer in hyper-encoder corresponds to the hyper-latents $\hat{z}$. The output of the last layer in decoder corresponds to the generated RGB image $\hat{x}$. The hyperprior feature and context feature are concatenated and fed into two "Linear" layers to yield the parameter, mean and deviation of a Gaussian distribution $\mathcal{N}(\mu, \sigma^2)$, for each latent. The output of the linear network must have exactly twice as many channels as the latents.

## A.5 TRAINING DETAILS

We implement all our models with PyTorch(Paszke et al., 2019). We use the Adam optimizer (Kingma & Ba, 2014) with $\beta_1 = 0.9$, $\beta_2 = 0.999$, $\epsilon = 1 \times 10^{-8}$, and base learning rate= $1 \times 10^{-4}$. When training transformers, it is standard practice to use a warmup phase at the beginning of learning, during which the learning rate increases from zero to its peak value (Vaswani et al., 2017). We use a warmup with 0.05 proportion of the total epochs. And then the learning rate decays stepwise for every 1/5 proportion epochs by a factor of 0.75. Gradient clipping is also helpful in the compression setup, which is set to 1.0. No weight decay is used in our models because rate constrain is inherently a regularization. We initialize the weights of our Entroformer with a truncated normal distribution.

Inspired by the self-supervised pre-training of Transformer, we also perform a random mask for pretrain. Different from *masked patch prediction*, masked latents are predicted solely by hyperprior while other latents are predicted by hyperprior and context. To do so we corrupt 50% of latents by replacing their value with 0. After pretrain, we finetune the models with regular manner.

All models are trained for 300 epochs with a batchsize of 16 and a patch size of $384 \times 384$ on 16GB Tesla V100 GPU card. To obtain models for different bit rate, we adapt the hyper-parameter $\lambda$ to cover a range of rate-distortion tradeoffs.

