# OpenReview forum: "Entroformer: A Transformer-based Entropy Model for Learned Image Compression"
_ICLR.cc/2022/Conference — ICLR 2022 Poster_

### Official Review · Reviewer_3XoD · 2021-10-20

**Correctness:** 4
**Technical Novelty And Significance:** 3
**Empirical Novelty And Significance:** 3
**Recommendation:** 8
**Confidence:** 3

**Main Review:**

Strenghts:
1) First approach using Transformer based method to image compression task.
2)  The proposed method outperforms previous methods based on CNNs.
3) An efficient parallel architecture is proposed and is more time-efficient than the serialized one on modern GPU device.

Weaknesses
Experiments could be more extensive.

**Summary Of The Paper:**

The authors propose a transformer-based entropy modeling order to capture long-range dependencies in probability distribution estimation. This model is optimized for image compression. The authors extend this architecture with a parallel bidirectional context
model to speed up the decoding process.

**Summary Of The Review:**

The paper present a novel contribution by using first Transformer based method to image compression task. Experiments validate the performance of the proposed approach named Entroformer compared to CNNs based methods.

---

> ### Author Response · Authors · 2021-11-19
> **Response to review 3XoD**
>
> Many thanks for your review.
>
> Following your comments, more experiments are provided in the appendix to justify the effectiveness of our model, such as
>
> 1) PSNR (RGB) on kodak dataset with more methods.
>
> 2) MS-SSIM on kodak dataset.
>
> 3) Comparison on weight with entropy model variants.
>
> 4) Performance of position encoding for varying the image size.
>
> Also, we provide more details about model architecture and training process in the appendix Sec. A.4, Sec. A.5.

---

### Official Review · Reviewer_JhCm · 2021-10-31

**Correctness:** 3
**Technical Novelty And Significance:** 2
**Empirical Novelty And Significance:** 2
**Recommendation:** 6
**Confidence:** 4

**Details Of Ethics Concerns:**

No ethical concerns.

**Main Review:**

I have not seen transformers used as entropy models for image compression before this paper (though there appears to be other submissions ICLR 2022 exploring the same idea). Transformers seem like a natural fit for this problem so an empirical evaluation is useful for the community. The diamond-shaped relative position encoding scheme introduced in this paper is also novel (as far as I know) and makes sense given the falloff in spatial correlation typical to natural images.

The use of a checkerboard decomposition for the latent tensor to create a "parallel bidirectional context model" is not novel (the authors appropriately cite He et al. 2021). The authors do provide runtime information (Table 2) showing that "serial context" models can be quite slow, while the parallel approach is much faster (170ms vs. thousands of milliseconds, though simpler models allow for decoding in under 22ms with the trade-off being worse compression rates).

My main concern with this paper is that the empirical evaluation does not show better results compared to the best published models. For example, Figure 5 compares again Qian (2021) but not He (2021), Minnen (2020), or Cheng (2020), all of which are cited elsewhere in the paper. It's hard to compare curves by eye across papers, but it appears that all of these methods have similar rate-distortion (RD) results. Furthermore, other models yield better RD results, for example:

Zongyu Guo, Zhizheng Zhang, Runsen Feng, Zhibo Chen. Causal Contextual Prediction for Learned Image Compression. IEEE Transactions on Circuits and Systems on Video Technology. 2021.

Changyue Ma, Zhao Wang, Ruling Liao, Yan Ye. A Cross Channel Context Model for Latents in Deep Image Compression. 	arXiv:2103.02884. 2021.

It appears (again, I'm eyeballing but you can trace the graph or contact the authors for more accurate numbers) that the Entroformer model hits 33.1 dB at 0.4 bpp, while Guo (2021) is closer to 33.5 dB and Ma (2021) is around 33.3 dB.

I do think that the entroformer model has an advantage since it allows for much faster decoding speeds (both Guo and Ma use a serial context model), but this speed benefit comes from the parallel decoding introduced by He (2021), not by the transformer introduced in this paper.



**Summary Of The Paper:**

This paper addresses the problem of learned image compression using a transformer as the entropy model. The authors introduce a diamond-shaped relative position encoding scheme that makes sense for image modeling. They also adopt a two-step, bidirectional context model based on a checkerboard-style spatial decomposition of the latent tensor.

The modeling choices are well-supported by an ablation study. In particular, the authors show that the checkerboard model is almost as good as a much slower spatial context model that is often used for learned image compression. They also show that the transformer-based entropy model ("entroformer") leads to a model with better rate-distortion performance compared to earlier models.

**Summary Of The Review:**

I think this paper is "marginally above the acceptance threshold" since the use of transformers as entropy models for image compression is novel and interesting. The paper needs a more thorough empirical evaluation to warrant a higher score.

---

> ### Author Response · Authors · 2021-11-19
> **Response to review JhCm**
>
> Many thanks for your review and we appreciate your insightful feedback.
>
> ● Comparison with the sota methods.
>
> Due to space limitation, in main text we compare our method with the most closely related deep learning methods. A more comprehensive SOTA comparison is appended in Appendix Sec A.1 Figure.11. Please let us know if there was any omission.
>
> ● Trade-off between speed and performance.
>
> A better trade-off between speed and performance is also a focus in our work. Compared to the CNN-based accelerated method (He et al., 2021), our transformer-based method can utilize rich context and achieve a better balance between speed and performance ( 3% bpp better).

---

### Official Review · Reviewer_FHR4 · 2021-11-04

**Correctness:** 3
**Technical Novelty And Significance:** 2
**Empirical Novelty And Significance:** 3
**Recommendation:** 6
**Confidence:** 5

**Main Review:**

** Strengths **

(1) The improvements over the CNN-based hyperprior and context models are promising.

(2) The ablation study justifies well the design choices.

** Weaknesses **

(1) Transformers are costly in terms of complexity and runtimes as shown in Table 2 (Transformer Serial). Some prior works introduce the non-local attention module (NLAM) to the hyperprior codec without using layers of transformer encoders. One example is "End-to-End Learnt Image Compression via Non-Local Attention Optimization and Improved Context Modeling", TIP2021. It is unclear how the proposed transformer-based hyperprior compares with such NLAM-based hyperprior design.

(2)  It is expected that transformer-based design may not scale up easily from the complexity perspective when it comes to processing higher-resolution images. Essentially, its complexity may grow exponentially with the spatial resolution of the image latents.

(3) In some ablation studies, the performance differences are expressed in BPP or delta BPP. It is unclear at what PSNR (or quality) level these numbers are reported. Why not report BD-rate savings and present a comparison of complete RD curves? The same argument applies to Figs. 6 & 7.

(4)  The number of network weights spent on the entropy coder should be compared.

**Summary Of The Paper:**

This work introduces a transformer-based entropy coding model for learned image compression. The backbones of the commonly used hyperprior encoder and decoder are replaced with transformer encoder layers, and the context model is replaced with the transformer decoder. The striking features of the proposed method include (1) a diamond-shaped relative position encoding with clipping, (2) a top-k selection scheme, and (3) a parallel bi-directional context model. The parallel bi-direction context model is a derivative work from He et al. 2021.

**Summary Of The Review:**

Overall, the idea is novel and interesting. The results look promising. However, there are some critical details missing. Further clarifications are needed to understand better the pros and cons of the transformer-based scheme as compared to the prior work.

---

> ### Author Response · Authors · 2021-11-19
> **Response to review FHR4**
>
> Thanks for the detailed feedback and we will further explain the question raised in the comment.
>
> ● Compared to non-local attention module.
>
> First, our Entroformer is purely based on Transformer block, while NLAM uses attention module. Second, our Entroformer uses position encoding to represent spatial information. Spatial information is crucial for compression task. In our Entroformer, considering the distance between latents, spatial information is also learned along with content information by attention module.
>
> Table.1 in the paper shows the effect of position encoding for transformer-based image compression. By removing the relative positional encoding, Entroformater's performance degrades by 5.0% bpp. In attention-based method, position encoding is effective for performance.
>
> ● Complexity with image resolution.
>
> Our method is based on the vanilla transformer whose complexity grows quadratically against spatial resolution. There are at least two straightforward remedies to address this issue: 1) Apply the attention module within a spatial window; 2) Use linear approximations of the attention module in the transformer. By the time of submitting this work, linear complexity transformer is a fast evolving research direction. Our method can be easily combined with these recently proposed lightweight transformers for better computational efficiency.
>
> ● Ablation study.
>
> Thanks for catching this. The PSNR level of the point has been reported in the caption of the tables and figures. For these ablation studies, we followed the configuration of previous work (Minnen 2018). In some configurations, there does appear to be slight variation in bpp. It would degrade readability when present RD-curves. We have experimented a lower rate point for the ablation of position encoding methods, which shows a consistent result. The PSNR of this point is ~29.1, and the bpp results are,
>
> CNNs:                                0.1724
>
> Trans w\o PE:                     0.1711(-0.8%)
>
> Trans with 2D APE:              0.1703(-1.2%)
>
> Trans with 1D+1D RPE:         0.1640(-4.9%)
>
> Trans with 2D RPE:         0.1611(-6.6%)
>
> Trans with 2D Diamond RPE: 0.1589(-7.8%)
>
> Due to the time constraint, we did not test at more points. The complete experiment with more testing points will be appended in our final version.
>
> ● The number of network weights.
>
> The comparison of network weights with other entropy models has been added to Appendix, Sec. A.4, Table 6.

---

### Decision · Program_Chairs · 2022-01-20

**Decision:**

Accept (Poster)

**Comment:**

The three reviewers all felt the paper was above threshold for acceptance to ICLR. To improve the final version, they suggest some additional discussion and experiments may help the paper.